# Reinstating motivational states: Electrical signatures of craving and neural mind reading

**Alice Mado Proverbio** [ID]*, **Alice Zanetti**

Cognitive Electrophysiology Lab, Department of Psychology, University of Milano-Bicocca, Milan, Italy

* mado.proverbio@unimib.it

## Abstract

The aim of this electroencephalogram (EEG) study was to identify electrical neuro-markers of 12 different motivational and physiological states such as visceral craves, affective and somatosensory states, and secondary needs. Event-related potentials (ERPs) were recorded in 30 right-handed participants while recalling a specific state upon the presentation of an auditory verbal command incorporating an evocative sound background consistent with that state (e.g., the chirping of cicadas associated with the verbal complaint about feeling hot). ERP data showed larger amplitude N400 responses in the affective and somatosensory states, while the P400 component displayed greater amplitudes for the secondary and visceral states. Furthermore, the two components were also discernibly responsive to the 12 micro-categories (e.g., joy vs. pain or hunger), by providing a distinctive electric pattern for mostly all microstates. The reconstruction of the intracranial generators of surface signals revealed common imagery-related activations, including the middle and superior frontal gyri, the fusiform and lingual gyri, supramarginal, and middle occipital regions, as well as the middle temporal region. Additionally, specific regions were identified that were active for distinct mentally represented content, such as that visceral needs were associated with activations in the medial and inferior frontal gyri, uncus, precuneus, and cingulate gyrus. Affective states were associated with activations in the medial frontal, superior temporal, and middle temporal gyri. Somatosensory states (e.g., pain or cold) activated regions in the parietal cortex and the crave for music was linked to activations in the auditory and motor regions. These findings support the use of ERP markers for BCI applications.

## Introduction

The aim of the present electroencephalographic (EEG) study was to identify neural markers associated with the mental representation of distinct motivational and physiological states, including visceral needs, affective and somatosensory experiences, and secondary goals. Rather than directly inducing such states — which would be

**Data availability statement:** All data generated or analyzed during this study are included in this published article and its supplementary information files. The stimulus set and experimental procedure, provided to ensure replicability, are available in the open repository: Proverbio, Alice Mado; Zanetti, Alice (2025). An auditory-mediated communication paradigm for evaluating individual needs and motivational states in locked-in patients. Bicocca Open Archive Research Data, V1. https://doi.org/10.17632/f2mg2n6cwv.1.

**Funding:** ATE – Fondo di Ateneo No. 31159-2019-ATE-0064, University of Milano-Bicocca.

**Competing interests:** The authors have declared that no competing interests exist.

both ethically unacceptable and methodologically unfeasible — we employed a guided mental imagery paradigm in which participants vividly recalled or simulated specific conditions (e.g., thirst, hunger, pain, joy) in response to auditory-verbal cues accompanied by evocative soundscapes congruent with each state. This approach allowed us to investigate whether distinct neural signatures could be detected and classified for each internally represented state, thereby advancing the possibility of developing EEG-based markers for affective and motivational decoding, with potential applications in brain–computer interface (BCI) systems for non-communicative patients [1–4].

Research on the neural foundations of mental imagery reveals a substantial overlap with perceptual processing, suggesting that imagination and perception share cognitive resources and neural substrates [5,6]. Dijkstra et al. [7] demonstrated that internally generated images resemble real percepts, albeit in attenuated form. Building on this, Dijkstra et al. [8] argued that differentiating perceptually driven experiences from self-generated imagery is intrinsically difficult, as both engage similar neural networks that converge at later processing stages. This convergence supports the "mental imagery emulation theory", which posits that imagination not only reproduces perceptual content but also mimics perceptual mechanisms themselves. Neuroanatomically, frontotemporal networks—crucial for working memory and attentional control—are central to imaginative activity [9]. Supporting this view, Dijkstra et al. [10] identified a top-down pathway from the inferior frontal gyrus (IFG) to the occipital cortex, active during imagination but not during perception, suggesting increased attentional demands in generating mental imagery.

Temporal dynamics further distinguish perception from imagination while revealing shared processes. Using MEG, Dijkstra et al. [11] showed that imagination lacks the sequential activation typical of perception, exhibiting slower and more diffuse neural engagement. Nevertheless, overlapping responses around 300 ms after stimulus onset suggest convergence at later stages, reflecting shared higher-order functions.

The neural bases of mental imagery involving emotional and motor components have received increasing attention. ERP studies reveal that imagining emotionally charged stimuli elicits enhanced late positive potentials (LPPs), particularly for negative content [12], indicating greater neural effort. Ogawa and Nittono [13] further linked imagery tasks to anterior N400 and centro-parietal N700 components. In the motor domain, EEG and ERP studies demonstrate distinct markers of imagined movement. Syrov et al. [14] found that N200 and slow potentials (SPs) differentiate between mental counting and motor imagery, with stronger and delayed SPs in the latter. Similarly, Mencel et al. [15] reported that reaching versus grasping imagery engages frontal and centro-parietal areas, aligning with Syrov et al. [14]. Collectively, these findings indicate that premotor and sensorimotor cortices are crucial to the mental simulation of motor actions.

Emotion-related imagery recruits neural networks overlapping with those engaged by real affective experiences. According to Lang's bio-informational theory, imagining an emotional event activates associative networks similar to those of direct perception. Ji et al. [16] showed that imagery can evoke both emotional and physiological

responses, enabling anticipation of affective outcomes. Costa et al. [17,18] found that imagining positive scenarios activates the medial prefrontal cortex (mPFC), nucleus accumbens, and amygdala, whereas negative imagery increases amygdala but reduces mPFC activity. However, because the mPFC is also implicated in working memory and arousal [19], its activation may not be uniquely tied to positive affect.

Neural and electrical markers of states such as sleep, hunger, craving, and emotion offer further insights. Sleep deprivation reduces N1 and P300 amplitudes in sensory and frontal cortices [20], implying diminished stimulus detection [21]. Hunger and thirst engage the anterior cingulate cortex and insula [22,23], while food-related stimuli evoke larger P1, P2, P300, LPP, and N200 amplitudes, particularly under deprivation [24]. Craving for substances or gambling activates prefrontal, cingulate, and fusiform regions [25] and modulates ERP components such as N170, P2, and P300 [26]. Motor desire involves deflections in cortical potentials ~400 ms before movement, implicating frontal, parietal, and motor areas [27], as well as increased theta activity in the SMA and dorsolateral PFC [28]. Neuroimaging corroborates these findings, revealing overlapping activation during real and imagined movement [29].

Desires linked to culturally mediated behaviors, such as gaming or music, show comparable neural signatures. Gaming craving enhances delta, theta, and beta activity in central and parieto-occipital areas [30], while social gaming desire activates temporal cortices [31]. Music listening engages reward circuits, including the nucleus accumbens and orbitofrontal cortex, and triggers dopamine release [32,33]. Positive emotions enhance activity in occipital and orbitofrontal regions [34], whereas fear modulates early ERP components (N1, P2) in centro-parietal areas [35], and sadness increases N2 and LPP amplitudes over occipital regions [36]. Pain perception likewise elicits enhanced N2 and P2 components [37]. Together, these findings demonstrate that motivation, sensory, and emotional circuits collectively underlie imagery and affective states.

ERP components serve as reliable neural markers, facilitating the study of perception and imagery and informing Brain-Computer Interface (BCI) development [38,39]. For instance, [40,41] identified distinct ERP signatures during visual and auditory imagery—centroparietal positivities (CPP), anterior negativities (AN), anterior positivities (AP), and P300-, N400-, or PN300-like responses—associated with specific categories such as faces, animals, music, or affective vocalizations. While imagery and perception share core components, imagery responses tend to be longer-latency and more anterior. In a subsequent study, [40,41] analyzed imagined motivational states (e.g., hunger, thirst, sleep, happiness, fear, play, pain), finding larger N400 amplitudes during perception than imagination, except for somatosensory states. The LPP was generally right-lateralized and similar across conditions. Differences in anterior N400 activity distinguished micro-categories (e.g., cold vs. heat, happiness vs. fear). These findings are promising for "mind-reading" BCI applications, particularly as deep learning algorithms have been developed to classify EEG/ERP patterns automatically [1,2,42,43].

In the present study, brief auditory commands were used instead of pictograms to test whether ERP modulation depends on the semantic nature of motivational states rather than visual features. We aimed to identify universal electrical markers generalizable across sensory modalities. Based on prior pictogram-based results [40], we hypothesized that both macro- and micro-motivational states would modulate late ERP components (P/N400) in distinct ways. Consistent with Dijkstra's framework [7,10], we expected imagery to elicit predominantly anterior activation beyond 300 ms, reflecting higher-order cognitive processing related to mental representations.

## Methods and procedure

### Participants

Thirty participants (17 females and 13 males) aged between 18–28 years, with no current or history of psychiatric or neurological disorders, took part in the study. Seven participants were subsequently excluded due to excessive EEG artifacts. The final sample included 23 participants, comprising 16 females and 7 males, with a mean age of 22.52 years (SE = 2.13) and an average education level of 16.3 years (SE = 1.72). All participants provided informed written consent and were

right-handed according to the Edinburg inventory. They were recruited through the online SONA System platform and received ECTS credits for their participation.

The research project, entitled "Auditory imagery in BCI mental reconstruction" was pre-approved by the Research Assessment Committee of the Department of Psychology (CRIP) for minimal risk projects, under the aegis of the Ethical committee of University of Milano-Bicocca, on February 9th, 2024, protocol n: RM-2024-775). Participant recruitment and testing started on 01/03/24 and ended on 10/06/24.

## Stimulus and materials

Auditory stimuli were designed by combining a human expressive voice with a background sound to evoke a context related to the targeted needs. The stimuli included: primary or visceral needs (hunger, thirst, and sleep), homeostatic or somatosensory sensations (cold, heat, and pain), emotional or affective states (sadness, joy, and fear), and secondary needs (desire for music, movement, and play). 17 audio clips were recorded for each microcategory, each replicated twice: once with a male voice and once with a female voice, totaling 408 stimuli. *Audacity* software was used to combining the vocal track with a background context coherent with the verbal content. Human voices were recorded using Microphone K38 by *Hompower* (SNR = 80 dB). The semantic content, the prosodic intonation and the emotional tone of all voices were coherent and appropriately matched. Some of the background sounds were recorded using the same microphone, while others were sourced from the publicly accessible BBC Sound Effects library for scientific purposes (https://sound-effects.bbcrewind.co.uk/search).

Stimuli were balanced for both length (number of words in the spoken message) and duration (in seconds) across the classes. Length measures were analyzed using a one-way ANOVA across the 12 micro-categories, which yielded a non-significant result [($F_{(11,176)}$ = 0.192, $p$ = 0.55], indicating that sentence lengths did not differ across categories. Stimuli intensity was balanced, with audio volume standardized to 89 dB using *MP3Gain Express*.

## Stimulus validation

The stimuli were validated using an independent sample of 40 participants (20 males, 20 females; M = 26.15 years, SE = 4.96), drawn from the same statistical cohort as the EEG study participants. The 408 audio recordings were distributed across two validation questionnaires, each tailored to the gender of the narrating voice. Each questionnaire comprised 204 acoustic stimuli, with 17 items per subcategory. The purpose of these questionnaires was to assess the efficacy and plausibility of the audio clips in conveying their respective motivational states. The surveys were conducted using the Qualtrics online platform and disseminated via invitation links shared in WhatsApp groups at the University of Milano-Bicocca. Participants were instructed to use headphones and were tasked with listening to each audio clip, identifying the described motivational state, and evaluating the plausibility of the content and the extent to which the voice effectively expressed the associated need. Responses were recorded on a 3-point Likert scale ranging from "not very plausible" to "very plausible." Each questionnaire required approximately 25 minutes to complete.

The questionnaire analyses demonstrated a remarkably high accuracy in identifying the content of the audio clips, with an overall mean rate of 96.33% (SE = 1.3) for correct responses. Notably, the highest accuracy rates were observed for audio stimuli representing the subcategories of thirst (98%), sleep (98%), cold (98%), happiness (98%), and fear (97%). Conversely, slightly lower accuracy rates were recorded for subcategories such as play (96%), music (96%), sadness (96%), pain (96%), hunger (95%), heat (95%), and movement (94%). Based on the validation data, only audio stimuli correctly identified in at least 70% of responses were included in the final stimulus set. Consequently, seven audio clips were excluded: four featuring a female voice and three a male voice. These excluded stimuli belonged to the subcategories of thirst (1 audio), heat (3 audios), sadness (2 audios), and movement (1 audio). On average, participants rated the acoustic stimuli as "very plausible" in 47.58% (SE = 7.13) of responses, "somewhat plausible" in 33.75% (SE = 3.33), and "not very plausible" in 18.67% (SE = 4.6). A statistical analysis was conducted to examine variance based on two factors:

"motivational states" (the four macro-categories) and "plausibility" (with levels: "not very plausible," "somewhat plausible," and "very plausible"). The analysis revealed a significant main effect of "plausibility" ($F_{(2, 808)}$ = 348.75, $p < 0.0001$; $\varepsilon = 0.879$, GG corr. $p$ value = 0.00001; $\eta_p^2 = 0.46$) and a significant interaction between the two variables [$F_{(6, 808)}$ = 9.75, $p < 0.00001$; $\varepsilon = 0.879$, GG corr. $p$ value = 0.00001; $\eta_p^2 = 0.07$]. The distribution of responses across the three plausibility levels differed significantly. Specifically, "not very plausible" was selected significantly less often than both "somewhat plausible" and "very plausible" (both $p < .001$). Furthermore, "very plausible" ratings were assigned significantly more frequently than "somewhat plausible" ($p < .001$). Post-hoc comparisons for the interaction revealed that somatosensory states were rated as "very plausible" significantly more often ($p < 0.001$) and as "not very plausible" significantly less often ($p < 0.05$) than the other macro-categories. These findings suggest that "somatosensory" auditory cues—representing states such as pain, heat, and cold—were perceived as the most realistic overall.

A repeated-measures ANOVA was conducted to examine significant differences between micro-categories and perceived plausibility. The factors analyzed included motivational states (12 micro-categories) and the "plausibility" variable, categorized into three levels: "unlikely," "somewhat likely," and "very likely." The analysis revealed a significant main effect for the plausibility [$F_{(2, 792)}$ = 3.69, $p < 0.001$; $\varepsilon = 0.893$, GG corr. $p$ value = 0.00001; $\eta_p^2 = 0.48$] and a significant interaction between motivational states and plausibility levels [$F_{(22, 792)}$ = 5.71, $p < 0.001$; $\varepsilon = 0.893$, GG corr. $p$ value = 0.00001; $\eta_p^2 = 0.14$]. Post-hoc analyses identified significant differences between specific micro-categories and plausibility levels ($p < 0.01$). The "hunger" micro-category was perceived as the least plausible, while the "cold" motivational state received the highest mean number of "very likely" responses and the lowest mean number of "unlikely" responses. Among affective states, "joy" elicited fewer "very likely" judgments, as did the secondary category of "movement desire."

## Experimental procedure

To ensure at least 50 EEG trials per micro-category, a subset of stimuli—specifically those with fewer classification errors and higher perceived realism—were repeated, resulting in 600 stimuli across 12 categories. These audio stimuli were organized into 18 sequences, each containing 33 or 34 stimuli. Due to the rapid presentation rate, sequences were structured in consecutive blocks of 4 or 5 items from the same micro-category to aid participant recall and imagery. Care was taken to avoid repeating any micro-category more than five times and to ensure a balanced representation of male and female voices. The order of micro-categories varied across sequences, and each sequence concluded with an attention-check question referencing to previously heard stimuli. Participants began by reading and signing an informed consent form and completing a laterality questionnaire. Subsequently, a 128-electrode EEG headset was applied, with the preparation process requiring approximately 25 minutes. During this time, participants reviewed the experimental instructions and completed a training session on imagery (Table 1). The training involved closing their eyes for 10 seconds to imagine an emotional state or need—movement, thirst, warmth, or sadness—corresponding to one of the four macro-categories. Afterward, participants reported the strategies they employed to imagine the state and indicated whether they recalled past experiences or generated new mental content during the exercise.

The experiment took place inside an electromagnetically and acoustically shielded booth (Faraday cage). Participants were seated in a comfortable chair positioned 114 cm away from a screen located outside the booth and wore *Trust*-brand headphones to listen to the audio stimuli. During the recording session, participants were instructed to fixate on a central point—a glowing cross positioned at the center of the screen—while refraining from moving their eyes, blinking, swallowing, or making any other movements, all while maintaining a relaxed posture.

The experimental task involved listening to the auditory cues and recalling the corresponding motivational state, as vividly and realistically as possible, as if experienced in the first person. During the imagery task, a yellow frame appeared on the screen to which ERP signals were synchronized. At the end of each sequence, the participant was presented with a question related to the stimulation content (e.g., Has anyone mentioned wanting a nice plate of pasta with meat sauce?) in the center of the screen and instructed to answer "yes" or "no" verbally a few seconds later. The question served to

**Table 1. Examples of suggested exercises for imagery training.**

| |
|---|
| *Movement* |
| - I imagine myself running freely on the soccer field with my teammates |
| - I imagine being in a tight crowd and trying to move through it |
| *Thirst* |
| - I imagine myself dehydrated with a dry mouth, yearning for an icy bottle of water on a sweltering day |
| - I go to the vending machine, grab a bottle of refreshing sparkling water, and finish it immediately |
| *Heat* |
| - I imagine myself in a hot environment, with clothes clinging to my body as I use a fan to cool down |
| - I imagine myself feeling unbearably hot and fanning my face with my hands. |
| *Sadness* |
| - I imagine myself sitting on a bench, or on the ground quite depressed, looking down, hiding my head, and withdrawing into myself. I am not paying attention to my surroundings |
| - I imagine not wanting to get out of bed, feeling down, with no desire to do anything |

verify the participant's actual execution of the task and attentional engagement. Each stimulus sequence began with three warning signals ("ATTENTION," "READY," and "GO"), displayed for 1 second in capital letters, and ended with "THANK YOU!" The words appeared in white on a gray background, in Times font, approximately 3 cm in size. Sequence presentation was randomized across participants, with consistent screen brightness and audio volume. Luminance conditions were scotopic. Each audio stimulus lasted 2500 ms, followed by a brief 100 ± 20 ms interval (ISI), after which a yellow frame appeared for 2500 ms. A 60 ± 20 ms interval (ITI) preceded the next audio stimulus (see Fig 1). Each item, consisting of

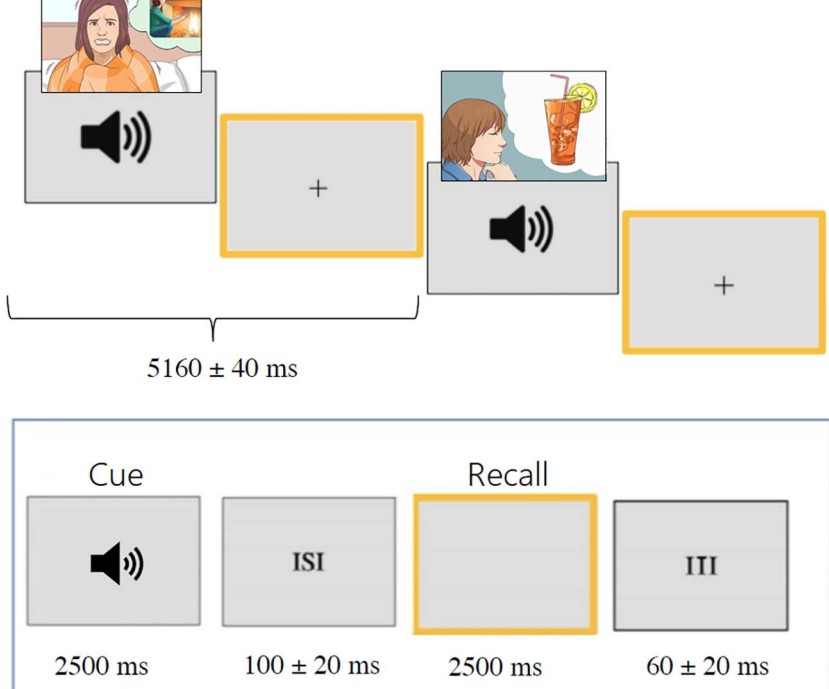

**Fig 1. Timeline of the experimental procedure.**

audio and imagery task, lasted 5160 ± 40 ms. Overall, each sequence lasted about 3 minutes, with brief pauses, and the entire experiment took approximately 1.5 hours. The auditory sequences were designed to ensure both the vivid evocation and the temporal stability of each motivational or physiological representation. To this end the same motivational state was presented in four or five consecutive repetitions, resulting in a total duration of approximately 21–26 seconds for each motivational state. This temporal design allowed participants sufficient time to form and maintain a vivid mental representation before the next condition began. In addition, to prevent semantic interference or emotional carryover, the order of presentation was fully randomized with the constraint that opposite or conceptually related states (e.g., hunger and thirst, joy and sadness) were never presented in close temporal proximity. A total of 18 sequences were presented, preceded by a shorter practice sequence. After the EEG recording, participants completed a Likert-scale questionnaire (1 = very difficult, 5 = very easy) assessing the ease of recall of motivational states related to primary needs, physical sensations, secondary needs, and emotions. The stimuli were presented using *Eevoke v2.2* (ANT Neuro).

### EEG recording and analysis

Brain activity was monitored using 128 electrodes placed according to the 10−5 International system, with horizontal and vertical electro-oculograms also being recorded. The averaged mastoids were used as the reference. Electrode impedance was kept below 5 kΩ, and the sampling rate was 512 Hz. EEG and EOG signals were captured using the *Cognitrace* system (ANT Software) and amplified with a bandpass filter (0.16–70 Hz). Any EEG artifacts exceeding ±50 μV were automatically rejected before averaging. EEG epochs synchronized with stimulus presentation were processed through the *EEProbe* system (ANT Software). ERPs were averaged offline from 100 ms before to 1200 ms after stimulus onset (of prompt for imagery).

The mean area amplitude of the anterior N400 component, measured between 400–600 ms at electrodes AF3, AF4, F7, and F8, and of the centroparietal P400, measured between 400–600 ms at electrodes CP1, CP2, and CPZ, was calculated. The choice of the electrode and the time window was based on where and when the components reached their maximum amplitude on the scalp, and on the previous literature. Measurements were conducted on individual ERP signals averaged across the 12 micro-categories and 4 macro-categories during the imagery task. The mean area amplitudes of the various components were subjected to repeated-measures ANOVA, with the following factors: motivational state (4 levels in the macro-category analysis and 12 levels in the micro-category analysis), electrode (depending on the component), and hemisphere (left, right). Using ASA software (ANT Software, Enschede, The Netherlands), characteristic topographical maps for the 4 macro-categories were generated, enabling visualization of the spatial distribution of the primary ERP components identified in the study: N400 and P400. These maps were created by plotting isopotential lines onto a color scale, obtained by interpolating voltage values between surface electrode sites. Overall. N400 component exhibited larger amplitudes at anterior regions, while P400 component was more pronounced at centroparietal sites.

### Source reconstruction

To identify the cortical sources of surface electrical activity in the P/N400 time window four swLORETA models were conducted on mean ERP averages corresponding to each macro-category of motivational state. *Low-Resolution Electromagnetic Tomography* (LORETA) is a powerful source reconstruction technique able to localize neural activity with high spatial resolution by estimating the sources of electrical activity within the brain [44]. In this study we used and advanced algorithm proposed by Palmero-Soler et al. [45] called SwLORETA, which incorporates a *Singular Value Decomposition* (SVD) based lead field weighting. Additionally, synchronization tomography and coherence tomography based on SwLORETA were introduced to analyze phase synchronization and standard linear coherence, applied to current source density. Both the head model's segmentation and generation were executed using Advanced Neuro Technology, a software program developed by ASA [46]. The swLORETA analysis yielded results indicating the statistical activation of a subset comprising

1056 dipoles. These results provide information on both their Magnitude values (in nA) and spatial coordinates following the Talairach & Tournoux [47] system. A dipole, in this context, represents the potential difference between the basal and apical portions of a cell, resulting in a positive and negative difference with an orthogonal orientation to the magnetic field. Utilizing the Collins dipole table each electromagnetic dipole was associated with its respective Gyrus and Brodmann Area. Source space properties were: Grey matter; grid spacing = 5 mm; estimated SNR = 3; Leadfield Normalization: Weighted.

## Results

### Self-reported data

**Ease of recall.** Data were collected from the whole sample of 30 students (13 male, 17 female), with an average age of 22.4 years (SE = 1.98). Mean scores were subjected to a one-way ANOVA featuring factor motivational state (4 levels of variability). Results indicated significant differences [$F_{(3,87)}$ = 8.25; $p < 0.00007$; $\varepsilon = 0.82$, GG corr. $p$ value = 0.0002; $\eta_p^2 = 0.22$]; post-hoc comparisons revealed that "visceral needs" (i.e., hunger, thirst and sleep) were significantly more easily imagined (M = 4.40; SE = 0.10) than other motivational states: "somatosensory" (M = 3.73; SE = 0.18) with $p < 0.05$; "secondary" (M = 3.47; SE = 0.16) with $p < 0.001$, and "affective" (M = 3.33; SE = 0.24) with $p < 0.001$.

**Stimulation content.** The percentage of correct responses to the final test (i.e., the final question about the run content) was 91.90% (SE = 7.04), corresponding to approximately 1.52 errors out of 18 questions (SE = 1.24). This indicates that participants paid close attention to the auditory stimulus semantic content as required for the imaginative task.

### Electrophysiological results

**Anterior N400 component (400–600 ms). Macro-categories.** The ANOVA performed on N400 mean area amplitude values recorded at AF3, AF4, F7, F8 sites in the 400−600 ms time window showed the significance of "state" factor [$F_{(3, 66)}$ = 3.10; $p < 0.03$; $\varepsilon = 0.892$, GG corr. $p$ value = 0.038; $\eta_p^2 = 0.13$], with larger N400 amplitudes during "affective" (M = −0.40; SE = 0.67) than "visceral" (M = 0.23; SE = 0.51) ($p < 0.05$) and "secondary" states (M = 0.21; SE = 0.58) ($p < 0.01$) as can be appreciated from ERP waveforms of Fig 2 and topographical maps displayed in Fig 3. The weak interaction of "state" x "electrode" [$F_{(3,66)}$ = 2.69; $p < 0.05$; $\varepsilon = 0.875$, GG corr. $p$ value = 0.07; $\eta_p^2 = 0.11$] and relative post-hoc comparisons showed that N400 was larger ($p < 0.001$) to "somatosensory" (M = −0.48; SE = 0.51) than "visceral" (M = 0.22; SE = 0.40) and "secondary" states (M = 0.26; SE = 0.44) at anterior frontal sites. The largest N400 response ($p < 0.01$) was recorded during recall of "affective" states at inferior frontal sites (M = −0.432, SE = 0.54),

**Micro-categories.** The ANOVA performed on N400 mean area amplitude values recorded in association with micro-states yielded the significance of "state" factor [$F_{(11, 242)}$ =2.25; $p < 0.05$; $\varepsilon = 0.6$, GG corr. $p$ value = 0.038; $\eta_p^2 = 0.093$]. Significances at post-hoc comparisons, along with N400 mean amplitude values for each of the microstates, and standard deviations, are reported in Fig 4.

Post hoc tests indicated that "joy" was the motivational state marked by the highest N400, in contrast to "social play," which was characterized by the lowest N400 amplitude. Fig 5 (left) shows the grand-average ERPs recorded in association with the 12 micro-categories at anterior and inferior left and right frontal electrodes, and at lateral and midline centro-parietal electrodes (right), separately for each reference microstate.

The study also examined whether a correlation existed between the ease of recalling motivational states, expressed in Likert scores ranging from 1 to 5, and the mean area amplitude of N400 responses recorded during each microstate. A Spearman's Rho correlation test was conducted on the two data sets, showing a linear correlation: the greater the difficulty in recalling a state, the larger the N400 amplitude ($r = 0.50612$; $p < 0.05$), as can be appreciated in Fig 6. This finding likely suggests that the N400 component reflected the cognitive effort required to evoke or recall motivational states.

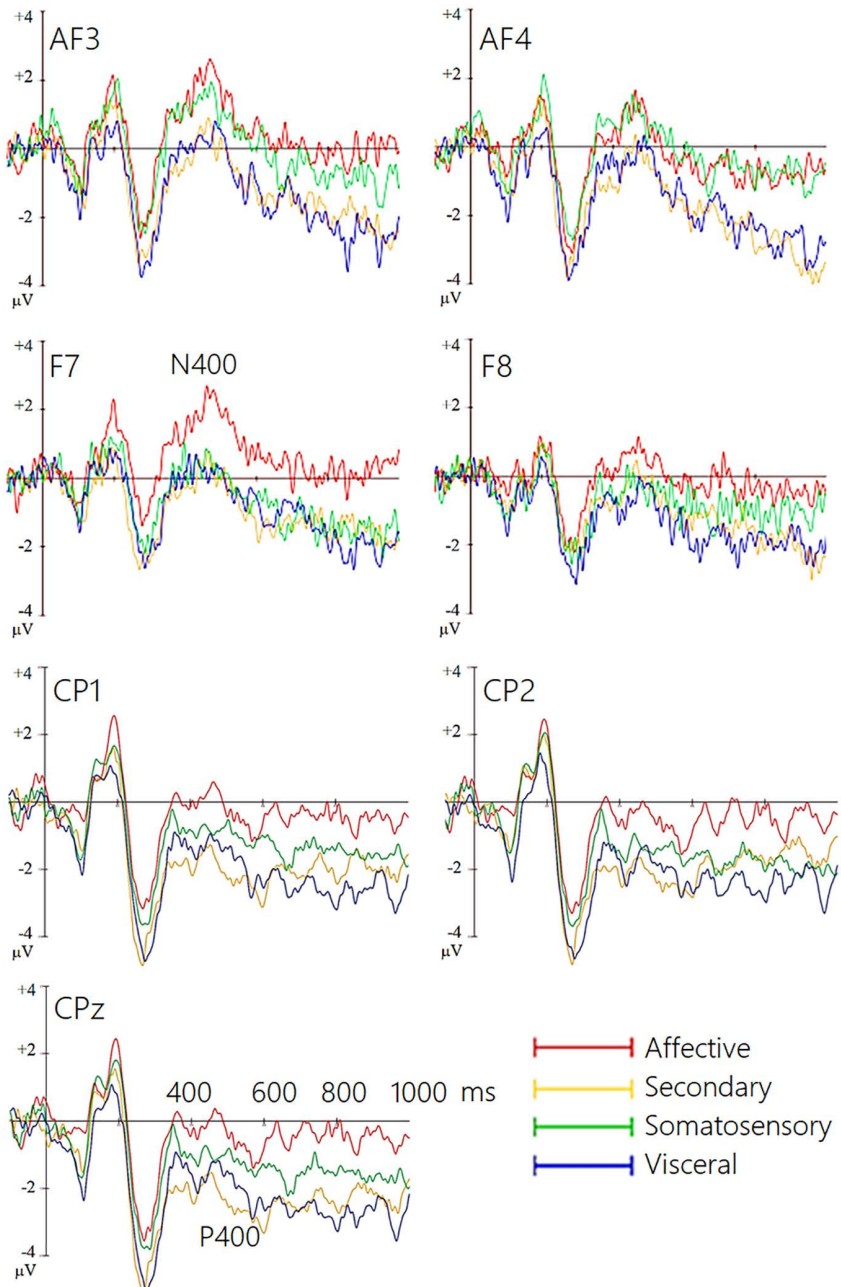

**Fig 2. Grand-average ERP waveforms recorded from left and right anterior and inferior frontal (top) and left, right and midline centroparietal sites (bottom) during recall of the 4 macro-categories of motivational states.**

**Centroparietal P400 response. Macrocategories.** The ANOVA performed on P400 mean area amplitude values recorded at CP1, CP2 e CPZ sites in the 400–600 ms time window showed the significance of "state" factor [$F_{(3,66)}$ = 2.84; $p < 0.05$; $\varepsilon = 0.7$, GG corr. $p$ value = 0.05; $\eta_p^2 = 0.11$]: post-hoc comparisons showed larger P400 responses during recall of "secondary" (M = 1.13; SE = 0.61) ($p < 0.008$) and "visceral" (M = 0.92; SE = 0.75) ($p < 0.03$) than "affective" states

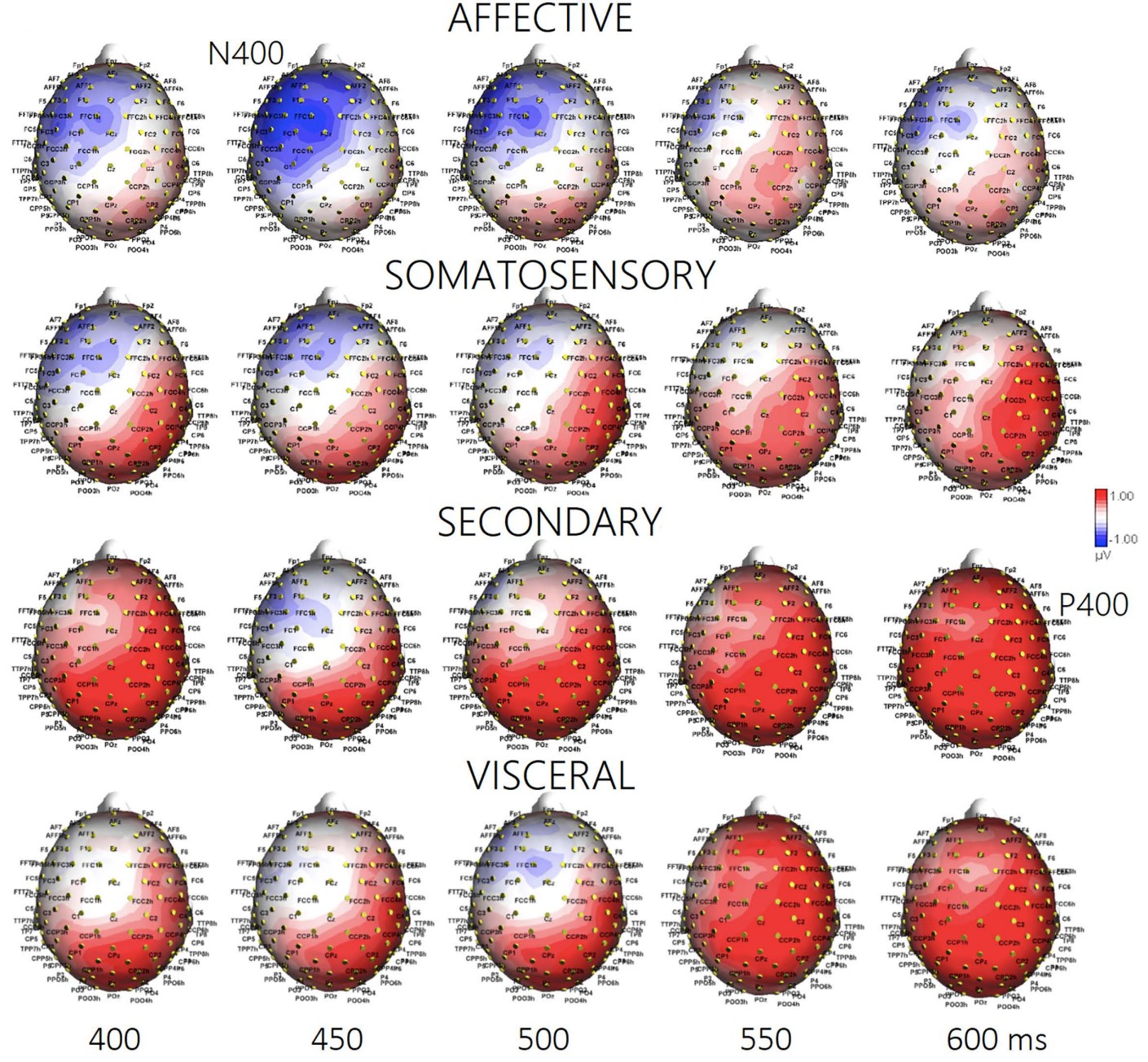

**Fig 3. Isocolour topographical maps of voltage potentials recorded during recall of the 4 macro-categories of motivational states in the 400-600 ms time window (50 ms step).**

(M = 0.23; SE = 0.59), with intermediate amplitude during recall of "somatosensory" states (M = .60; SE = .74), see Fig 2 (Lower) and 3 for ERP waveforms and topographical maps.

**Microcategories.** The ANOVA performed on P400 amplitude values recorded in association with microstates yielded the significance of "state" factor [$F(11, 242) = 2.05$; $p < 0.025$; $\varepsilon = 0.7$, GG corr. $p$ value = 0.05; $\eta_p^2 = 0.10$]. Significances at post-hoc comparisons, along with P400 mean amplitude values for each of the microstates, and standard deviations, are

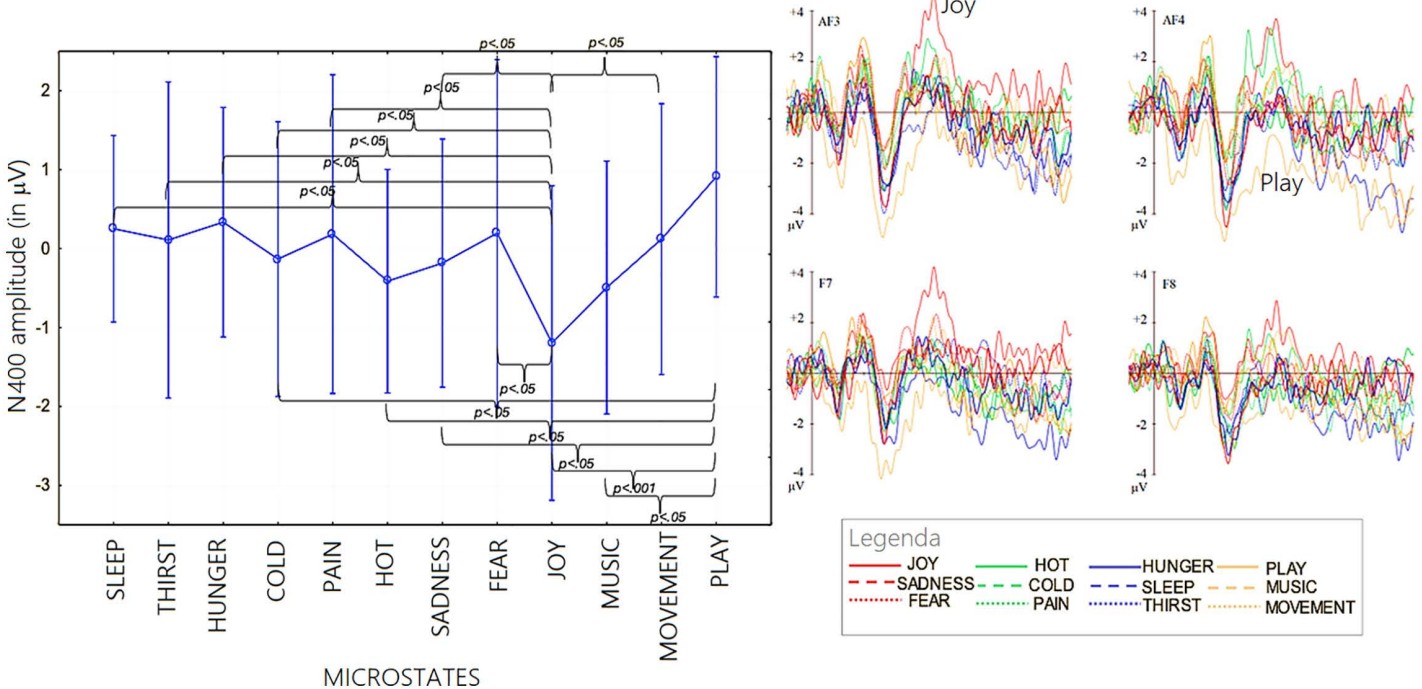

**Fig 4. (Left) Mean area amplitude of N400 response recorded as a function of the 12 micro-categories of motivational states.** Standard deviations and statistical significances at post-hoc comparisons are also shown for the various states. (Right) Grand-average ERP waveforms recorded from left and right anterior and inferior frontal sites during recall of the 12 microstates.

reported in Fig 7. Post hoc tests indicated that "play" ($p < 0.0002$) and "movement" ($p < 0.0006$) were the motivational state marked by the highest P400, in contrast to "joy" and "fear" which were characterized by the lowest P400 amplitude. Fig 5 (right) shows the grand-average ERPs recorded in association with the 12 micro-categories at anterior and inferior left and right frontal electrodes, and at lateral and midline centroparietal electrodes, separately for each reference microstate. It can be concluded, therefore, that the amplitude of the ERP component P400 recorded at centro-parietal sites changed depending on the mentally represented micro-category.

The correlation between P400 mean amplitude values and the ease in recalling motivational states was not significant. This suggests that imaginative effort or efficacy was expressed more by N400 responses.

**swLORETA source reconstruction.** In Table 2 are listed the active electromagnetic dipoles significantly explaining surface potentials within the 400–600 ms time window during the recall of the four motivational states according to swLORETA [45]. Almost all macrostates featured the activation of the left superior frontal gyrus (BA10) involved in Self-representation and working memory (inner voice), as well as posterior brain regions involved in Imagery. Affective motivational states (Table 2a) specifically involved right fronto-temporal areas, namely the right medial frontal (MFG), superior temporal, and middle temporal gyri (MTG). Additionally, several other neural areas traditionally associated with visual imagery were activated, including the fusiform, lingual, and supramarginal gyri. Fig 8 displays the localization of more active areas as projected in sagittal, coronal and axial MRI sections.

Table 2b lists the more active dipoles during the recollection of "secondary" desires, which included the right MTG, STG (BA38) and the bilateral limbic areas (BA20 and Ba28) possibly linked to musical imagery, and left precentral gyrus (BA4) linked to motor imagery [48] (see neuroimages in Fig 8).

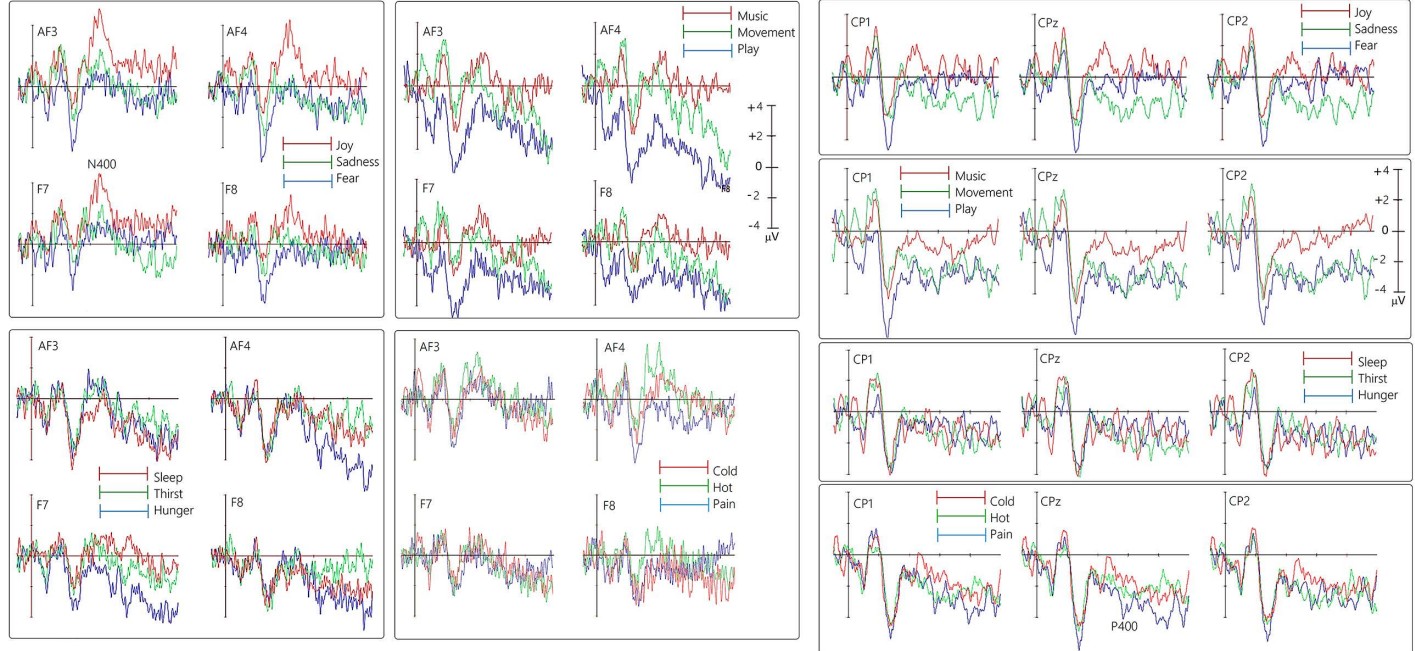

**Fig 5. Grand-average ERP waveforms recorded from left and right anterior and inferior frontal (left) and left, right and midline centroparietal sites (right) during recall of the 12 micro-categories of motivational states.**

In Table 2c, the areas involved during the recall of "somatosensory" sensations are listed. The most distinctive areas appeared to be the angular (BA39) and postcentral gyri on the right (BA1), as well as the left and right inferior parietal lobes (BA40) (see Fig 8). Table 2d lists the areas involved in the subjective recall of visceral needs. The most active areas included the right MFG and IFG, right precuneus, left and right uncus involved in craving and hunger, besides imagery related areas (e.g., right lingual gyrus and left middle occipital gyrus), as shown in Fig 8.

**ROI analysis.** In order to have a comprehensive view of the specific sources of electro-magnetic activity across the 4 motivational states, 8 regions of interest (ROIs) per hemisphere were identified following the ROI clustering procedure used to perform statistical analyses on individual LORETA solutions [49–52]. The selected ROIs are listed in Table 3.

The source reconstruction data enabled the identification of the most active Regions of Interest (ROIs, in nA) during the recall of various motivational macrostates. Fig 9 visually depicts the most engaged ROIs in the right and left hemispheres (in terms of total strength of active ROIs), highlighting a distinct pattern that lends itself well to potential classification using deep learning techniques. A 2 way ANOVA applied to average magnitudes computed as a function of hemisphere and ROI (as within factors) and macro-category (as between factor) yielded the significance of Hemisphere factor [$F(1, 28) = 8.97$, $p < 0.006$; $\eta_p^2 = 0.24$], with larger signals over the right (RH = 11.01; SE = 1.83) than left (LH = 5.03, SE = 0.87 µV) hemisphere. The further interaction of Hemisphere x ROI [$F(7,24) = 2.74$; $p < 0.03$; $\varepsilon = 0.7$, GG corr. $p$ value = 0.045; $\eta_p^2 = 0.44$] and relative post-hoc comparisons showed stronger right-sided signals especially over temporal, fusiform and orbitofrontal ROIs.

## Discussion

This study investigated ERP correlates associated with 12 motivational states elicited through mental representations, organized into four macro-categories. Significant differences emerged across both micro- and macro-categories, affecting late

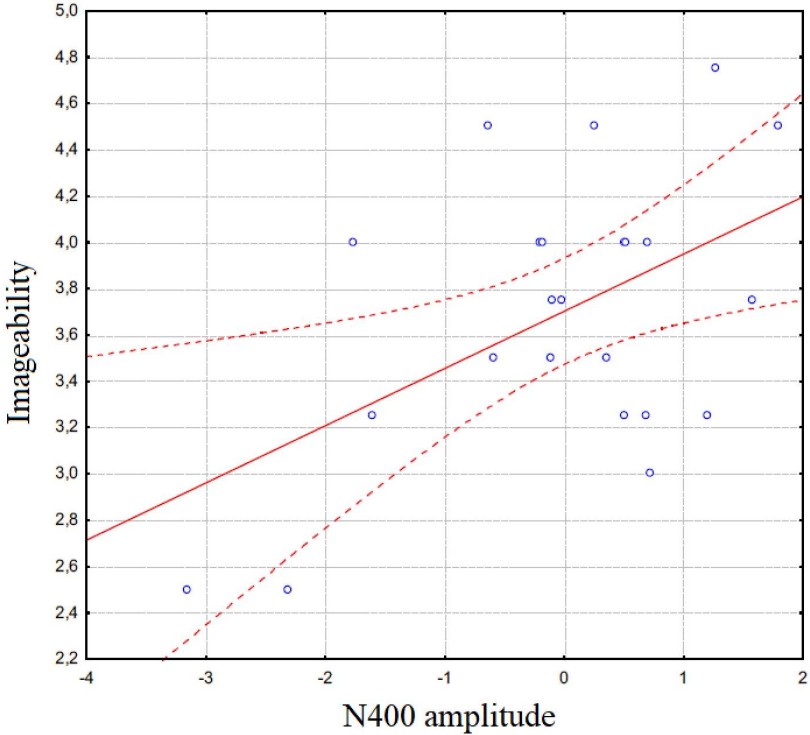

**Fig 6. Spearman Rho correlation between subjective rates of complexity of recalling the motivational states and individual N400 mean amplitude values.**

latency components (N400 and P400). A for previous studies with a similar paradigms [e.g., 39,41], no category-specific effects on sensory or perceptual components (i.e., N170, N2, P2, or P300) was observed during recall of imagined states. Indeed, early ERP overlaps reflect processing of the identical bright frame (the visual cue with which ERP epochs were synchronized), with no differences attributed to category-related imaginative processes earlier than 300 ms [11,38].

*Component-Specific Effects* Similar to the ERP study in which motivational states were prompted using pictograms [40], here an anteriorly negative N400 component with a 400–600 ms latency, as well as a centro-parietal positive P400 within the same time window, were identified. However, unlike the current study's auditory cues, the pictogram-based recall elicited a later positive component (LPP with an 800–1000 ms latency), likely due to the additional processing time required to interpret visual pictograms compared to auditory cues [53]. In both studies, the N400 component localized primarily to anterior frontal and inferior frontal regions, suggesting a common neural mechanism for mentally activating motivational states. Frontal and fronto-temporal areas are known to play a pivotal role in imaginative processes, specifically in initiating, controlling, and sustaining mental representations [9,10]. The N400 amplitude was notably greater when recalling "affective" states, particularly joy, as well as "somatosensory" states, such as pain and cold, paralleling findings from the previous pictogram-based study. Overall, brain activity in the P/N400 latency stage was larger over the right hemisphere, particularly for over temporal, fusiform and orbitofrontal ROIs: this finding fits with previous data by [40,41] that similarly demonstrated increased right hemisphere involvement during imagery tasks, particularly for the LPP component, and other literature on mental imagery [54–56].

The N400 component appears to be highly sensitive to cognitive effort related to motivational states. Conceptually, larger N400 amplitudes, particularly in anterior regions, reflect greater cognitive processing when recalling or distinguishing states with high emotional salience, such as "affective" states, indicating an increased mental load associated with

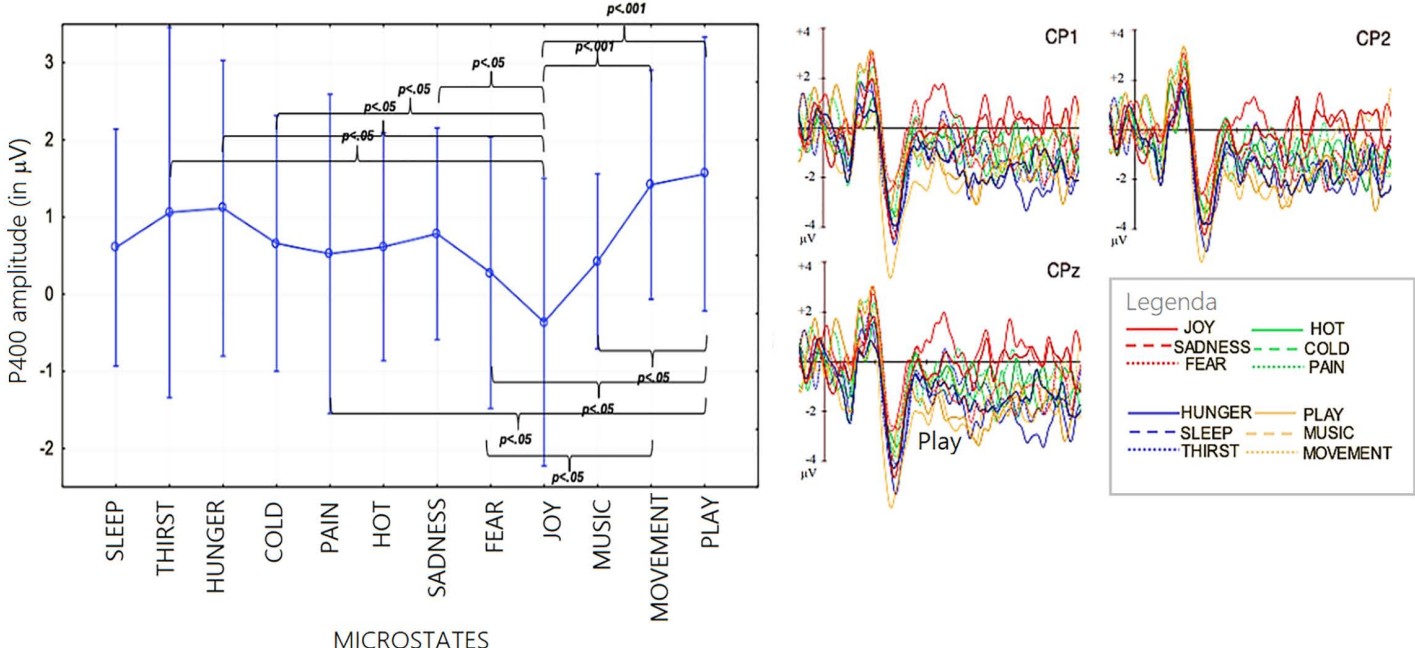

**Fig 7. (Left) Mean area amplitude of P400 response recorded as a function of the 12 micro-categories of motivational states.** Standard deviations and statistical significances at post-hoc comparisons are also shown for the various states. (Right) Grand-average ERP waveforms recorded from left, right and midline centroparietal sites during recall of the 12 microstates.

emotionally charged memories. For micro-categories, N400 variations suggest that distinct motivational nuances (e.g., "joy" versus "social play") require different degrees of cognitive engagement, with heightened N400 amplitudes linked to more challenging or intense states. The correlation between N400 amplitude and the subjective difficulty of recall further supports the role of N400 as an index of cognitive effort in retrieving complex or personally significant motivational experiences. The P400 component, more prominent at centroparietal sites, reflects categorical distinctions in motivational representation, particularly among different types of internal states (e.g., "secondary" or "visceral" versus "affective" states). While less sensitive to recall difficulty, P400 amplitudes vary with specific micro-categories, indicating that this component may capture differences in the sensory or embodied aspects of motivational states, particularly for states involving movement and play, without directly indexing cognitive effort.

In sum, the N400 aligns with cognitive effort linked to emotional recall, while the P400 provides insight into the differentiation of motivational states based on their embodied characteristics. Overall the amplitude of the ERP component anterior N400 (see also [38]) and centro-parietal P400 significantly changed depending on the mentally represented micro-category. In this study source reconstruction of scalp-recorded potential was also carried out for "mind reading" purposes.

*State-Specific Activations* The swLORETA reconstruction of intracortical generators revealed consistent activations across several macrocategories uniquely linked to imagery. The left superior frontal gyrus was engaged during "affective" and "visceral" states, while the left middle frontal gyrus was predominant in "somatosensory" ones. These areas may sustain imagination and working-memory processes, as observed by [19] and Lima et al. [57], and their stimulation can even evoke visual hallucinations [58]. The superior frontal gyrus also contributes to self-representation—the "inner voice" [59]—and self-referential thought [60,61], essential for representing personal needs. The frontoparietal network has been implicated in episodic memory [62], whereas the vmPFC seems to orchestrate mental scene construction [63], since its damage impairs imagination unless externally cued.

**Table 2. List of active electromagnetic dipoles (along with their Talairach coordinates) explaining brain voltage during the four recalled macro-motivational and physiological states (group analysis).**

**a. AFFECTIVE STATES (Joy, Sadness, Fear)**

| Magn. | H | Lobe | Gyrus | BA | |
|---|---|---|---|---|---|
| 13.7 | LH | F | Superior frontal | 10 | Self & Working memory |
| 13.1 | RH | F | Fronto/polar | 10 | Affective processing, prospective memory |
| 8.64 | RH | T | Superior temporal | 38 | Negative emotions |
| 8.63 | RH | T | Middle temporal | 21 | Sadness |
| 6.93 | RH | F | Superior frontal | 6 | Attention, self-awareness, default-mode network |
| 6.91 | LH | F | Middle frontal | 6 | |
| 6.61 | LH | F | Superior frontal | 8 | |
| 6.56 | RH | F | Precentral | 6 | |
| 5.15 | RH | P | Postcentral | 1 | |
| 5.02 | RH | P | Superior parietal lobule | 7 | |
| 3.70 | LH | P | Supramarginal | 40 | |
| 4.93 | LH | T | Fusiform | 20 | Visual imagery and processing of social info |
| 4.21 | RH | T | Fusiform | 19 | |
| 4.17 | LH | T | Fusiform | 37 | |
| 4.14 | LH | O | Lingual | 17 | |
| 4.11 | RH | O | Lingual | 17 | |
| 3.61 | LH | O | Cuneus | 19 | |
| 3.38 | LH | O | Middle occipital | 18 | |

**b. SECONDARY STATES (Social play, Music and Movement)**

| Magn. | H | Lobe | Gyrus | BA | Function |
|---|---|---|---|---|---|
| 15.9 | RH | T | Fusiform | 20 | Social play |
| 14.8 | RH | T | Medial temporal | 21 | |
| 14.6 | RH | T | Fusiform | 20 | |
| 13.2 | RH | T | Superior temporal | 38 | Music recall |
| 12.7 | RH | Limbic | Uncus | 20 | |
| 9.46 | LH | Limbic | Uncus | 28 | |
| 10.4 | RH | F | Superior frontal | 10 | Attention, self-awareness, Default-mode network |
| 9.09 | LH | F | Medial frontal | 9 | |
| 9.20 | RH | P | Postcentral | 3 | Action and movement |
| 8.39 | LH | F | Precentral | 4 | |
| 7.89 | RH | F | Medial frontal | 6 | |
| 5.15 | RH | P | Superior parietal lobule | 7 | |
| 5.30 | LH | O | Medial occipital | 18 | Imagery |
| 4.89 | RH | O | Cuneus | 19 | |
| 4.72 | LH | T | Medial temporal | 21 | |
| 3.92 | LH | O | Cuneus | 18 | |
| | | | | | |

**c. SOMATOSENSORY STATES (Pain, Cold, Heat)**

| Magn. | H | Lobe | | BA | Function |
|---|---|---|---|---|---|
| 11.8 | LH | F | Superior frontal | 10 | Self & Working memory |

*(Continued)*

**Table 2.** (Continued)

**a. AFFECTIVE STATES (Joy, Sadness, Fear)**

| Magn. | H | Lobe | Gyrus | BA | |
|-------|-----|-------|-------|------|---|
| 10.3 | RH | F | Rostro/medial prefrontal | 10 | Introspection, self-awareness, default-mode network |
| 9.97 | RH | F | Medial frontal | 11 | |
| 9.63 | RH | T | Middle temporal | 21 | |
| 9.21 | RH | T | Fusiform | 37 | |
| 6.41 | RH | F | Superior frontal | 6 | |
| 6.19 | RH | F | Middle frontal | 46 | |
| 5.17 | LH | P | Inferior parietal lobe | 39 | Somatosensory processing |
| 3.99 | RH | P | Inferior parietal lobe | 40 | |
| 3.92 | RH | P | Angular | 39 | |
| 3.65 | RH | P | Postcentral | 1 | |
| 4.75 | LH | T | Middle temporal cortex | 21 | Imagery |
| 3.36 | LH | O | Middle occipital | 19 | |
| 3.07 | RH | O | Cuneus | 19 | |
| 2.39 | LH | O | Lingual | 18 | |

**d. VISCERAL STATES (Hunger, Thirst, Sleep)**

| Magn. | H | Lobe | Gyrus | BA | Function |
|-------|-----|-------|-------|------|----------|
| 11.2 | LH | F | Middle frontal | 10 | Self & Working memory |
| 10.7 | RH | F | Superior frontal | 10 | Attention, Imagery |
| 10.6 | RH | T | Inferior temporal | 20 | |
| 10.3 | RH | P | Postcentral | 3 | Gustatory |
| 8.00 | RH | Limbic | Uncus | 20 | Craving |
| 6.48 | RH | P | Precuneus | 19 | Visceral |
| 6.08 | LH | F | Inferior frontal | 45 | Gustatory |
| 10.0 | RH | F | Medial frontal | 10 | Attention, self-awareness, default-mode network |
| 9.99 | RH | T | Middle temporal | 21 | |
| 5.99 | RH | F | Medial frontal | 6 | |
| 5.86 | RH | F | Superior frontal | 6 | |
| 5.80 | LH | F | Precentral | 6 | |
| 5.28 | RH | Limbic | Cingulate | 31 | Craving |
| 5.25 | LH | Limbic | Uncus | 28 | Craving |
| 4.70 | LH | T | Middle temporal | 21 | |
| 4.58 | RH | O | Lingual | 18 | |
| 4.46 | LH | O | Middle occipital | 18 | |

Magn. = magnitude in nA; H = Hemisphere, BA = Brodmann areas, function = presumed functional properties. Shaded are the key structures identified as most distinctive for a BCI application.

The occipitotemporal cortex (BA18, 19, 21, 37) was active across all imagery types, consistent with visuo-spatial imagery findings [9,19,64,65]. Specific activations further differentiated motivational states: bilateral cuneus engagement appeared in all but "visceral" states, while the left MTG was selective for "secondary" and "visceral" ones. The cuneus, involved in visual imagery but inversely related to its vividness [66], and the lingual gyrus contribute to visual and mnemonic processes, from orientation to face recognition and language [67]. The left MTG supports mental representation and theory of mind [64,65,68,69], thus forming a neural bridge between perception, emotion, and imagination.

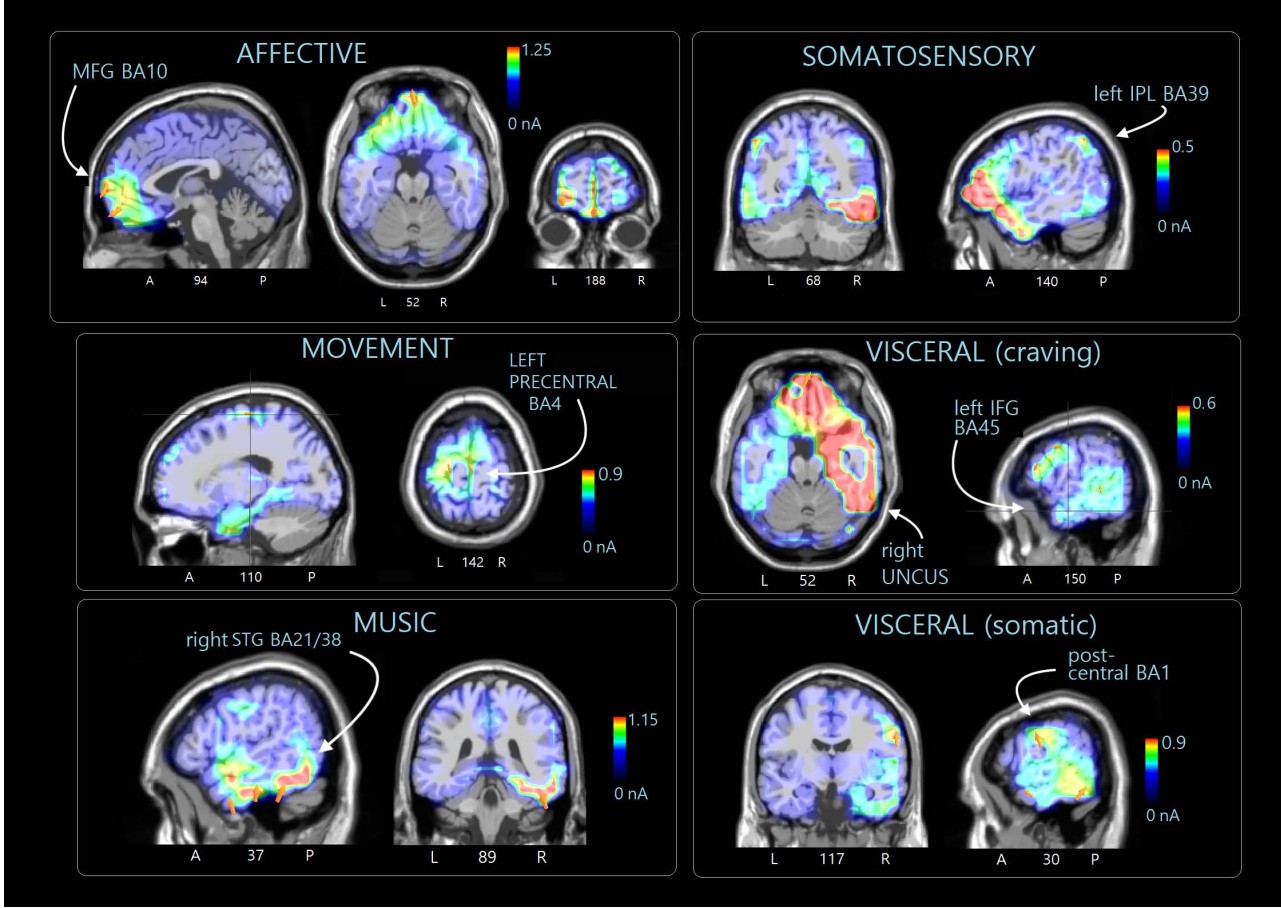

**Fig 8. Sagittal, axial and coronal views of swLORETA source reconstructions of N400 surface potentials recorded in the 400-600 ms time window during the various macro-categories of motivational states.** The various colors represent differences in the magnitude of the electromagnetic signal (nA). The electromagnetic dipoles appear as arrows and indicate the position, orientation and magnitude of the dipole modelling solution applied to the ERP waveform in the specific time window. A, anterior; P, posterior; L, left; R, right; numbers refer to the displayed brain slice in the MRI imaging plane.

Affective imagery engaged the right fusiform and left supramarginal gyri. The fusiform gyrus mediates face and body perception, whose impairment leads to prosopagnosia affecting both recognition and imagery [19]. The supramarginal gyrus (BA40), part of the DMN, contributes to self-awareness, mental and spatial imagery [70–72]. Affective recall also involved the right frontopolar cortex, associated with affective processing and prospective memory, similarly active during joyful imagery [34]. The orbitofrontal cortex (OBF), integral to reward circuits [73], has been linked to happiness [34,74] alongside the mPFC, PCC, and inferior parietal lobule. The right superior temporal gyrus may reflect the emotional and social nature of auditory stimuli [34], as corroborated by Gawda et al. [75], who found right MFG engagement for "joyful" words. Notably, the right MTG, also discussed by Proverbio et al. [34], is consistently tied to sadness and reduced in patients with affect-recognition deficits [76].

"Secondary" motivational states recruited the right fusiform and MTG, both linked to social play [31]. The right superior temporal gyrus, including Heschl's gyrus, was active during music imagery [40], consistent with studies of auditory perception and imagination [69,77–79]. Self-generated auditory imagery similarly recruits these cortices [80]. Movement-related

**Table 3. List of Regions of Interest identified and referenced to the Gyri and Brodmann areas (BAs) included in each cluster. A similar clustering was used in Della Vedova and Proverbio [31] and in Proverbio and Cesati [34].**

| ROIs | BA | GYRUS |
|------|-----|-------|
| OCC | 18, 19 | OCCIPITAL CORTEX: Inferior Occipital, Middle Occipital, Superior Occipital, Lingual (also BA 17) gyri, Cuneus (also BA 17) |
| FUSIF | 19, 20, 37 | FUSIFORM AREA: Fusiform Gyrus, Cerebellum (Anterior Lobe), Posterior lobe (Declive), Middle occipital (only BA 37) and Inferior temporal gyri (only BA 37) |
| TEMP | 19, 20, 21, 22, 38, 39, 41, 42 | SUPERIOR, MIDDLE AND INFERIOR TEMPORAL CORTEX: Superior Temporal, Middle Temporal and Inferior Temporal Gyri |
| PREM | 4, 6 | PREMOTOR CORTEX: Precentral (also BA 43), Middle Frontal, Medial Frontal, Superior Frontal, Precentral gyri, Paracentral Lobule |
| FRONTAL | 8, 9, 46 | MEDIAL, SUPERIOR FRONTAL AND DORSOLATERAL PREFRONTAL CORTEX: Middle Frontal, Medial Frontal and Superior Frontal Gyri |
| ORB\IF | 10, 11, 44, 45, 47 | ORBITOFRONTAL AND INFERIOR FRONTAL CORTEX: Superior Frontal, Medial Frontal, Middle Frontal, Inferior Frontal and Rectal Gyri |
| PARIETAL | 1, 2, 3, 7, 19, 39, 40 | PARIETAL CORTEX: Superior and Inferior Parietal Lobule, Precuneus, Postcentral, Supramarginal and Angular Gyri |
| LIMBIC | 20, 23, 24, 28, 31, 34, 35, 36, 38 | LIMBIC AREA: Uncus, Cingulate Gyrus, Anterior Cingulate, PPA |

"secondary" states engaged the left precentral gyrus [81] and right MFG, including the SMA [82] and parietal BA3–BA7, as also noted by Della Vedova and Proverbio [31] and Urgesi et al. [83].

"Somatosensory" states (cold, pain) activated bilateral inferior parietal lobes, right angular and postcentral gyri—areas central to both real and imagined tactile experiences [84]. Lesions of the inferior parietal lobe cause asymbolia for pain [85], and S1 [86] directly represents tactile and nociceptive imagery content.

Finally, "visceral" states (hunger, thirst) activated the right MFG and left IFG, linked to hunger, as well as the uncus, precuneus, and cingulate gyrus, involved in craving. The frontal operculum (BA45) mediates gustatory processing [87], responding more to ongoing taste than satiety [88]. The right MFG shows enhanced activity in eating disorders [89], though it also responds to sweet stimuli in healthy individuals [90]. The uncus and cingulate, both limbic, are modulated by stress-induced craving [91]. The precuneus, similarly active during craving and food-seeking [92], shows increased DMN connectivity with the dorsal cingulate and retrosplenial cortices in anorexia [93], perhaps reflecting a cognitive effort to suppress desire through enhanced self-monitoring.

*Theoretical Implications* On the basis of the reviewed literature, the regions identified as sources of EEG-recorded electrical activity appear broadly consistent with, and plausibly reflective of, the specific motivational states participants were instructed to recall. The evidence further suggests that imagination may recruit neural systems strikingly similar to those engaged during real, firsthand experience. In this study, distinct ERP components were associated with twelve motivational micro-categories—hunger, thirst, sleep, warmth, cold, pain, joy, sadness, fear, movement, music, and play—organized within four broader domains: visceral, somatosensory, affective, and secondary needs. Such differentiation underscores the neural specificity of motivational imagery. The identification of key ERP markers—most notably the N400, amplified for affective and somatosensory states, and the P400, more prominent in visceral and secondary ones—offers a promising avenue for decoding motivational states from neural signals.

These findings hold translational relevance for brain–computer interface research [e.g., 1], particularly in assessing consciousness and unexpressed needs in patients with severe disorders of consciousness.

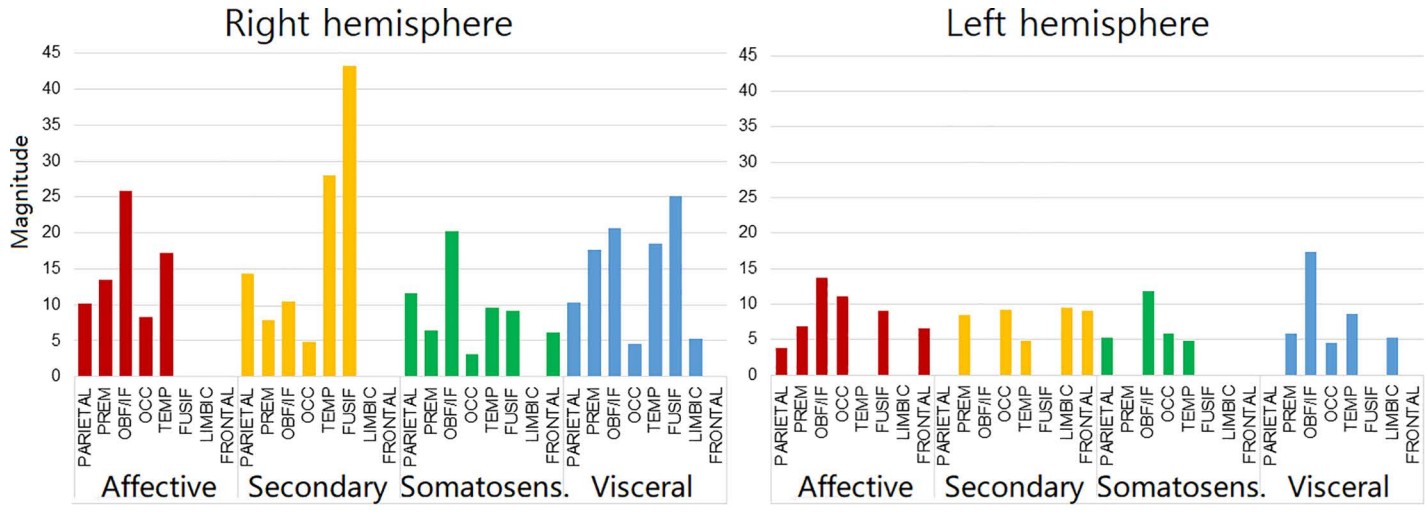

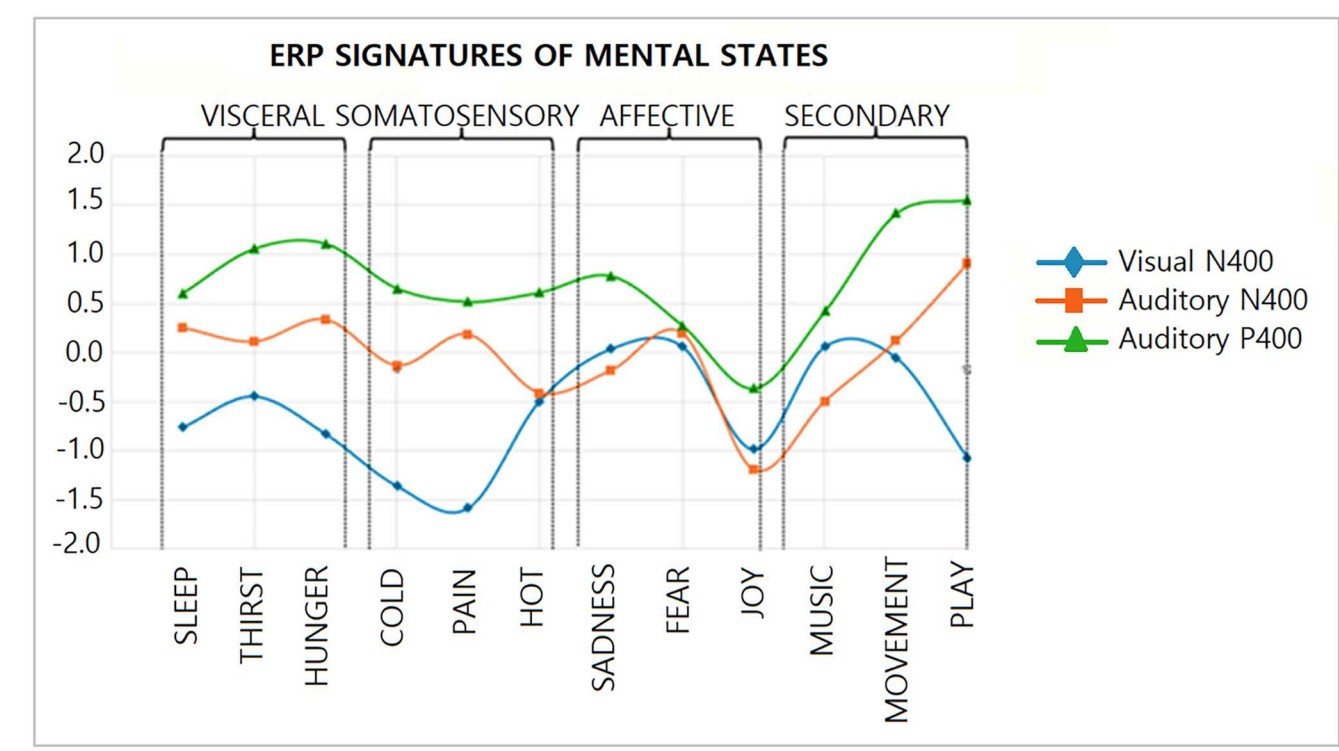

**Fig 9. (Top) Cumulative values of average magnitude (dipole strength in nA) of electromagnetic dipoles found active in each of the selected ROIs of the left and right hemisphere, during the recall of states belonging to the 4 macro-motivational states.** (Bottom) The amplitude trend profile of the P/N400 components recorded in the present, and in Proverbio & Pischedda's [34] studies, based on visual pictograms, shows a clear similarity in the behavior of these neural signatures, irrespective of the experimental condition.

## Study limitations and future directions

However, as with many studies in this area, our research faces several limitations. First, the small sample size and restricted age range limit the generalizability of the findings; expanding the sample size and including more diverse age groups could enhance the robustness and applicability of these results. Additionally, only twelve motivational states were examined, though investigating a broader array of motivational states could yield important insights into the needs patients affected by consciousness disorder. Future research should aim to broaden the participant pool across a wider age range, including minors, adults aged 28–65, and elderly individuals (65+). Expanding the study to encompass other motivational states within each macro-category, such as emotional states like anger or disgust, and secondary needs like wanting to hear a story or be cared for, may be especially relevant for understanding everyday needs. Lastly, applying this framework to clinical populations, such as visually impaired individuals, may reveal meaningful differences in brain activity, further enriching the potential applications of this research. A further potential limitation might come from the fact that the imaginary motivational states were to be voluntary activated, and did not derive from real homeostatic needs (such as hunger or drug craving). This condition may not fully correspond to people's experiences in real situations related to such needs, but the same criticality holds for any study involving imagery paradigms. We recognize that the spatial resolution of swLORETA is limited, although this constraint is unlikely to substantially affect the interpretation of our findings.

## Acknowledgments

We are grateful to Francesca Pischedda for her kind contribution to the experimental paradigm.

## Author contributions

**Conceptualization:** Alice Mado Proverbio.

**Data curation:** Alice Mado Proverbio, Alice Zanetti.

**Formal analysis:** Alice Mado Proverbio, Alice Zanetti.

**Funding acquisition:** Alice Mado Proverbio.

**Investigation:** Alice Mado Proverbio, Alice Zanetti.

**Methodology:** Alice Mado Proverbio, Alice Zanetti.

**Resources:** Alice Mado Proverbio.

**Supervision:** Alice Mado Proverbio.

**Writing – original draft:** Alice Mado Proverbio, Alice Zanetti.

**Writing – review & editing:** Alice Mado Proverbio.

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
