## [Decision Letter · Decision Letter 0]

6 Nov 2025

Dear Dr. Proverbio,

I received evaluations from two experts and, as you can see below, their opinions are conflicting: the first reviewer is very positive about your work, the second reviewer, on the other hand, is very critical on several points. I appreciate the viewpoint of both of them and thank them very much and therefore I decided to give you the opportunity to convince both reviewers in a major revision. I cannot guarantee that another round of revisions will lead to the publication of the manuscript, but I recommend that you focus on both theoretical (e.g., justifying the imaginative paradigm) and statistical aspects.

We look forward to receiving your revised manuscript.

Kind regards,

Giulia Prete

Academic Editor

PLOS ONE

Journal Requirements:

“ATE – Fondo di Ateneo No. 31159-2019-ATE-0064, University of Milano-Bicocca”

3. In the online submission form, you indicated that “All data generated or analyzed during this study are included in this published article and its supplementary information files. The data supporting the findings of this study are available from the corresponding author mado.proverbio@unimib.it upon request. Due to privacy restrictions, some data may not be shared publicly.”

Reviewers' comments:

Reviewer's Responses to Questions

**Comments to the Author**

1. Is the manuscript technically sound, and do the data support the conclusions?

Reviewer #1: Yes

Reviewer #2: Partly

2. Has the statistical analysis been performed appropriately and rigorously?

Reviewer #1: Yes

Reviewer #2: No

3. Have the authors made all data underlying the findings in their manuscript fully available?

Reviewer #1: Yes

Reviewer #2: Yes

4. Is the manuscript presented in an intelligible fashion and written in standard English?

Reviewer #1: Yes

Reviewer #2: Yes

Reviewer #1: Dear Editor,

I have carefully read and evaluated the manuscript titled "Decoding Motivational States and Craving through Electrical Markers for Neural 'Mind Reading’". The paper presents a highly innovative and valuable contribution to the fields of cognitive neuroscience and BCI. By integrating ecologically grounded auditory-evocative prompts with event-related potentials and source localization techniques, the authors elegantly demonstrate that distinct motivational and physiological states yield differential electrophysiological signatures. This fine-grained neuromarker-based decoding of mental states opens up fascinating new avenues for both theoretical modeling and potential translational applications.

Below, I provide specific comments and suggestions aimed at further refining and clarifying this already impressive piece of work:

Major Suggestions for Minor Revision:

Graphical Clarity in Figures 4 and 7

While the data visualizations are generally effective, I recommend improving the readability of the axis labels in Figures 4 and 7, particularly along the x-axis. These appear overly compressed or poorly contrasted, especially when compared to the more legible formatting seen in Figure 9. For consistency and accessibility, the same graphical standards should be uniformly applied across all figures.

Effect Size Reporting in ANOVAs

To enhance the statistical transparency and reproducibility of the findings, I strongly encourage the authors to report effect size metrics—specifically partial eta squared (η²) and epsilon (ε) values—for all ANOVAs performed. While significance testing provides valuable information, effect sizes offer a more nuanced understanding of the strength and practical relevance of observed differences. Given the complexity and richness of the ERP data across multiple states and components, effect size indices are essential for readers to assess the robustness and scope of the effects reported.

Subsectioning the Discussion

The Discussion would greatly benefit from clearer structure through the use of sub-paragraphs or thematic subheadings. Given the breadth of the results—ranging from component-specific findings (e.g., N400, P400) to source localization and state-by-state interpretations—organizing the discussion around these axes would improve clarity and reader engagement. Suggested subheadings might include: Component-Specific Effects, State-Specific Activations, Theoretical Implications, and Future Directions.

Minor Language and Style Adjustments

The manuscript is generally well-written and the scientific language is appropriate. However, I suggest minor improvements to fluency and idiomatic expression. For instance:

Page 10, line 287:

Instead of:

"Has it been mentioned (in this experimental sequence) to desire a nice plate of pasta with meat sauce?"

I suggest: "Has the notion of desiring a plate of pasta with meat sauce emerged at any juncture during the course of this experimental sequence?" This revised phrasing better aligns with the formal tone of the manuscript and enhances clarity.

Reviewer #2: Unfortunately, despite its strong sides, the study suffers from some conceptual and methodological weaknesses, which, in my opinion, prevent publication. Most important are pointed out below:

Aim is presented as detecting markers of different motivational and physiological states. However, the study uses imagery and does not induce these states. However, while the relationship between imagery and real states is covered in the introduction, the described aim of the study remains misleading.

The introduction seems to only partially cover the current literature, and sometimes the findings seem to be picked up from the literature pool to support a particular claim rather than present the existing views. For example, citing data showing that activation of mPFC is related to positive scenarios, while many studies support the role of this area in self-referential processes with no regard to valence.

I am also concerned about the number of different states that the Authors wanted to induce during the relatively long procedure. I doubt whether it was possible to keep participants really involved and deeply following such diverse and altering imagery states. Especially since these states are expected to induce specific bodily states.

The components of the evoked potentials are interpreted as responses for imagery. This is also seriously problematic for me, as the exact timing and intensity for imagery cannot be determined. I’m afraid that the discussed ERP components are the responses for a cue (frame) as the only well temporally defined stimulus. The Authors do not convince me that they observe imagery-evoked responses.

The parameters and settings of source localization are not reported at all. This is not trivial, as localization methodology is very susceptible to improper settings.

The final analysis includes only 23 participants, which is a small sample and does not keep up with the current standards. Especially facing the number of analyses (variables and their levels).

It would be beneficial to report not only anova differences between categories but also confusion matrices, as the aim of the study was focused towards mind reading.

**Do you want your identity to be public for this peer review?** For information about this choice, including consent withdrawal, please see our Privacy Policy

Reviewer #1: No

Reviewer #2: No

---

## [Author Response · Author response to Decision Letter 1]

10 Nov 2025

Please see the attached Rebuttal Letter and Letter to the Editor

---

## [Decision Letter · Decision Letter 1]

7 Dec 2025

Reinstating Motivational States: Electrical Signatures of Craving and Neural Mind Reading

PONE-D-24-53416R1

Dear Dr. Proverbio,

We’re pleased to inform you that your manuscript has been judged scientifically suitable for publication and will be formally accepted for publication once it meets all outstanding technical requirements.

Kind regards,

Giulia Prete

Academic Editor

PLOS One

Additional Editor Comments (optional):

One of the original Reviewer is unable to revise the present version of the manuscript, but I have carefully evaluated the revision and I am happy to endorse the publication of the manuscript in its present form.

Reviewers' comments:

Reviewer's Responses to Questions

**Comments to the Author**

Reviewer #1: All comments have been addressed

2. Is the manuscript technically sound, and do the data support the conclusions?

Reviewer #1: Yes

3. Has the statistical analysis been performed appropriately and rigorously?

Reviewer #1: Yes

4. Have the authors made all data underlying the findings in their manuscript fully available?

Reviewer #1: Yes

5. Is the manuscript presented in an intelligible fashion and written in standard English?

Reviewer #1: Yes

Reviewer #1: The authors have satisfactorily addressed all the concerns raised during the review process. The revisions are thorough, well-executed, and significantly strengthen the manuscript. I am pleased with the outcome and recommend the paper for acceptance

**Do you want your identity to be public for this peer review?** For information about this choice, including consent withdrawal, please see our Privacy Policy

Reviewer #1: No

---

## [Editor Report · Acceptance letter]

PONE-D-24-53416R1

PLOS One

Dear Dr. Proverbio,

I'm pleased to inform you that your manuscript has been deemed suitable for publication in PLOS One. Congratulations! Your manuscript is now being handed over to our production team.

Kind regards,

on behalf of

Dr. Giulia Prete

Academic Editor

PLOS One